# In Vitro: The Extraordinary Enhancement in Rutin Accumulation and Antioxidant Activity in Philodendron “Imperial Red” Plantlets Using Ti-Mo-Ni-O Nanotubes as a Novel Elicitor

**DOI:** 10.3390/biotech13030024

**Published:** 2024-07-04

**Authors:** Hanan S. Ebrahim, Nourhan M. Deyab, Basamat S. Shaheen, Ahmed M. M. Gabr, Nageh K. Allam

**Affiliations:** 1Department of Plant Biotechnology, Biotechnology Research Institute, National Research Centre (NRC), Cairo 12622, Egypt; hanansamir272@gmail.com; 2Energy Materials Laboratory (EML), School of Sciences and Engineering, The American University in Cairo, New Cairo City 11835, Egypt; nourhan.deyab@gmail.com (N.M.D.);; 3Physical Chemistry Department, National Research Centre (NRC), Cairo 12622, Egypt; 4Academy of Scientific Research and Technology, 101 Kasr El Ainy St. Kasr El Ainy, Cairo 11516, Egypt

**Keywords:** *Philodendron erubescens*, Ti-Mo-Ni-O nanotubes, novel elicitor, rutin, antioxidant activity

## Abstract

Rutin, a flavonoid phytochemical compound, plays a vital role in human health. It is used in treating capillary fragility and has anti-Alzheimer, anti-inflammatory, and antioxidant effects. In this study, Ti-Mo-Ni-O nanotubes (NTs) were used, for the first time, in an unprecedented plant biotechnology application, wherein in vitro Philodendron shoots (*Philodendron erubescens*) known as “Imperial Red” were targeted for rutin accumulation. The antioxidant responses and the accumulation of rutin were evaluated in treated *Philodendron erubescens* (*P. erubescens*) shoots using 5.0 mg/L of Ti-Mo-Ni-O NTs. The total phenolic content and total flavonoid content were estimated, and an ABTS^+^ assay, FRAP assay, and iron metal chelation assay were performed. The application of Ti-Mo-Ni-O NTs enhanced the rutin content considerably from 0.02 mg/g to 2.96 mg/g for dry-weight shootlet extracts. Therefore, the use of Ti-Mo-Ni-O NTs is proposed to be a superior alternative to elevate the rutin content. The aim of the current study is to target *P. erubescens* shoots grown in vitro for the accumulation of rutin compounds using Ti-Mo-Ni-O NT powder, to determine the quantitative and qualitative accumulation of rutin via HPLC–DAD analysis, and to estimate the antioxidant activity of *P. erubescens* shoot extract. This study presents a novel methodology for utilizing nano-biotechnology in the synthesis of plant secondary metabolites.

## 1. Introduction

The Philodendron is a type of plant that belongs to the family Araceae (commonly known as Arum), and while it was originally native to the Caribbean, Colombia, and Venezuela, it is now extensively cultivated in Asia. About ten of the hundreds of species have been designated as ornamental plants, including *Philodendron erubescens* “Imperial Red”. Furthermore, research shows that the plant contains triterpenoids and flavonoids, which have medicinal effects against a variety of diseases [1]. Most plant tissues contain phenolic compounds, which are secondary metabolites because they are not required for growth, development, or reproduction; nevertheless, they may have roles as antioxidants and in interactions with the plant’s biological environment. Rutin is a phytochemical flavonoid with a variety of therapeutic uses and has antioxidant, antibacterial, antiallergenic, anticancer, and anti-inflammatory capacities [2].

Different types of biotic and abiotic elicitors have been used for the elicitation of plant secondary metabolism. Nanomaterials have shown great potential to act as novel elicitors of flavonoids in plants due to their unique properties. Recently, different types of nanoparticles (NPs) have been used as effective elicitors of flavonoids in in vitro cultures of various plant species. Different types of NPs, including metallic NPs, metal oxides, and carbon-related NPs, are widely reported as novel elicitors of flavonoids. These NPs include silver (Ag) NPs, copper (Cu) NPs, iron oxide (Fe_3_O_4_) NPs, zinc oxide (ZnO) NPs, copper oxide (CuO) NPs, and chitosan-based NPs [3,4].

Nanotubes (NTs) are another class of nanostructured materials that have a hollow cylindrical shape with a diameter measuring on the nanometer scale and long lengths depending on the synthesis method. The large surface-area-to-volume ratio of the nanotubes can allow their performance to far exceed that of other nanostructures. In this regard, carbon nanotubes (CNTs) have been the most common nanomaterial employed as an elicitor of flavonoids [5,6]. However, investigations of nanotubes that are low in cost, easily synthesized, and have high performance are still scarce.

Titanium (Ti) is a beneficial element for plants. When administered through roots or leaves at low concentrations, Ti has been shown to improve agricultural performance by enhancing the activity of particular enzymes, boosting the chlorophyll content and photosynthesis [7,8], facilitating nutrient absorption, enhancing stress tolerance, and increasing crop yield and quality. The interactions of titanium (Ti) with other nutrients, notably iron (Fe), affect its function in plants. When plants lack iron, Ti helps to activate genes that help them obtain it, which enhances both plant growth and Fe uptake and utilization [8]. Titanium dioxide (TiO_2_) NPs not only enhanced shoot characterization but also improved phenolic compound (rosmarinic acid and caffeic acid) accumulation in *Mentha piperita* [9]. Another essential element for the development of plants is molybdenum (Mo) [10]. The microstructure of the roots and the activity of the rhizobia in the symbiotic system were discovered to be affected by Mo-based nanomaterials that enter plant tissues, which weaken the ability of the soybean–rhizobium symbiotic system to fix nitrogen [11]. Mo is necessary for the growth of most plants, like other metals, acting as a cofactor for particular plant enzymes such as coenzymes MoCo and FeMo to engage in plant reduction and oxidation [12]. In addition to plants, Mo is recognized as a crucial nutrient for bacteria and animals [13]; however, the only form of Mo found in bacteria and plants is the anion form, and an absence of Mo is fatal to living things [14]. Plants lacking Mo develop slowly and have little chlorophyll [15]. Because Mo is a crucial component of nitrate reductases and nitrifying enzymes, which may regulate the reduction in inorganic nitrate and support the proper N_2_ fixation of plants, previous research has demonstrated that Mo plays a significant role in the proper nitrogen fixation of plants [16]. It is an essential component of two enzymes that convert nitrate into nitrite (a toxic form of nitrogen) and then into ammonia before it is used to synthesize amino acids within the plant. It is also needed by symbiotic nitrogen-fixing bacteria in legumes to fix atmospheric nitrogen [11]. Because of its remarkable effectiveness in fixing nitrogen, the legume plant–rhizobium symbiotic nitrogen fixation system is a unique type of plant–microbe symbiosis found in the rhizosphere micro-ecosystem [17]. Nitrogenase, an enzyme complex with an Fe-Mo cofactor, is responsible for catalyzing biological nitrogen fixation in nature [18]. On the other hand, nickel oxide nanoparticles (NiO NPs) made via green synthesis utilizing a variety of plant extracts have been studied for their distinctive physiochemical characteristics and potential biological uses [19]. Green synthesis involves appropriate, more convenient, straightforward, and environmentally friendly synthesis processes to prevent the synthesis of irritating waste products and hazardous compounds. Therefore, research stresses the biogenic nano-sized nickel oxide’s antioxidant and antibacterial effects on the *Buxus wallichiana* plant, which possesses antioxidant terpenoids, polar, nonpolar, tannin, saponin, phenol, and ascorbic acid with pronounced potential to scavenge the function of free radicals [20]. Two Gram-positive and two Gram-negative bacterial strains were used to investigate the antioxidant and antibacterial capability of biogenic NiO NPs. Humans who are infected with *Bacillus* species experience headaches, vomiting, and diarrhea. In addition to spoiling milk, *Escherichia coli* can infect food. As a result, NiO NPs are biologically ready to resolve, stop, and manage these problems [21]. NiO NPs have the capacity to combat produced free radicals and reduce or oxidize a variety of species. They have drawn a lot of interest in recent years due to their exceptional electrical, optical, magnetic, catalytic, chemical, thermal, and mechanical characteristics [22]. 

Mixed and hybrid metal oxide nanomaterials have shown better performance than their single counterparts in many applications because of the synergetic interaction that balances the properties of each component, producing higher performance. For the first time in nano-biotechnology, we used Ti-Mo-Ni-O NTs as novel elicitors in plants. Rutin, a flavonoid phytochemical, has several beneficial effects on human health, including acting as an antioxidant, treating capillary fragility, preventing Alzheimer’s disease, and reducing inflammation. The aim of the current study is to target *P. erubescens* shoots grown in vitro for the accumulation of rutin compounds using Ti-Mo-Ni-O NTs powder, then determine the quantitative and qualitative accumulation of rutin by (HPLC–DAD) analysis and estimate the antioxidant activity for their *P. erubescens* shoot extract. 

## 2. Materials and Methods

### 2.1. Ti-Mo-Ni Oxide NTs Preparation 

The nanotube powder was prepared from Ti-0.3Mo-0.8Ni sheets via breakdown anodization [23]. The metal sheets were cleaned in acetone, ethanol, and deionized water in an ultrasonic cleaner for 15 min, sequentially. The anodization setup was made of a Ti-0.3Mo-0.8Ni sheet as a working electrode and graphite rod as a counter electrode dipped into 0.1 M of perchloric acid electrolyte solution at 0 °C and connected to a DC power supply at 13 V. The process stopped when the Ti-0.3Mo-0.8Ni sheet was transformed completely into fine white nanotube powder. The electrolyte solution was filtered to collect the powder, which was then washed thoroughly with deionized water and dried overnight. The nanotube powder was annealed in air at 450 °C for 4 h with a heating rate of 5 °C/min. Figure 1 shows the preparation steps to obtain the Ti-Mo-Ni oxide NTs powder. The morphology and the elements present in the powder were examined by Zeiss FESEM Ultra 60 microscope and an Energy Dispersive X-ray (EDX) unit attached to the Zeiss microscope. The crystalline structure was analyzed using a X’Pert PRO MRD diffractometer (Cu source at a scan rate (2°) of 0.05° s^−1^). 

### 2.2. Plant Material

The Philodendron “Imperial Red” (*Philodendron erubescens*) shoots were purchased from the Agronomy Research Institute, Agricultural Research Center (ARC), Giza, Egypt. *P. erubescens* shoots were started and maintained on MS [24] solid medium that included 0.5 mg/L of BA and Kinetin, 3% sucrose, and 8.0 g/L agar with pH of 5.7. Regenerated shoots were subculture once every 4 weeks on MS medium without plant growth regulators and used as plant materials to conduct this experiment.

### 2.3. Ti-Mo-Ni-O NTs Experiment on Philodendron erubescens

Around 5.0 mg/L of Ti-Mo-Ni-O NTs powder was added to the liquid MS medium before autoclaving [25]. After sterilization, sonication was used for an hour at 40 °C before culturing the Philodendron shoots. Cultured shoots were shaken at 90 rpm for three days. The effects of Ti-Mo-Ni-ONTs were investigated on the rutin accumulation in Philodendron shoots extract using HPLC–DAD analysis and antioxidant activities.

### 2.4. High-Performance Liquid Chromatography with a Diode-Array Detector (HPLC–DAD) Analysis

Agilent Technologies, Palo Alto, CA, USA, produced the Agilent 1100 series HPLC system, which was used for the HPLC determination and was outfitted with a quaternary pump G1311A, a degasser G1322A, a UV detector, and Agilent ChemStations Rev. B. 04.03. A LiChrospher RP-18 HPLC column (250 mm, 4.6 mm, 5 m; Merck, Darmstadt, Germany) was used for the separation. In the mobile phase, solvent A (water with 0.1% TFA) and solvent B were used (acetonitrile). The elution procedure was followed, with a flow rate of 1.0 mL/min from 0 min at 5% B to 12 min at 100% B. At room temperature, the analysis was performed in triplicate, with the detection wavelengths set to 250, 280, and 320 nm. Each methanolic extract had a 10 mg/mL concentration. The calibration curves were created using one mg of the standard rutin compound, which was precisely weighed, dissolved in methanol, and utilized in the concentration range of 7.0–500 µg/mL. 

### 2.5. Phytochemical Analysis

Two mg of the samples was combined with one mL of the methanol. The mixes underwent 30 min of sonication. For the microplate reader analysis, FluoStar Omega was used to record the outcomes.

#### 2.5.1. Total Phenolic and Total Flavonoids

Using the Folin–Ciocalteu technique described elsewhere [26], the total phenolic content was assessed. In a nutshell, the process involved combining 100 µL of diluted Folin–Ciocalteu reagent with 10 µL of sample or standard in a 96-well microplate. A total of 80 µL of 4 N Na_2_CO_3_ was then added, and the mixture was incubated for 20 min in the dark at room temperature (25 °C). The resultant blue complex hue was measured at 630 nm at the conclusion of the incubation period. Data are displayed as means and SDs. For the total flavonoids assay, the aluminum chloride technique, as reported elsewhere [27], was used to determine the total flavonoid content, with some small modifications made in microplates. In a 96-well microplate, 15 µL of sample and standard were first added. Then, 175 µL of methanol and 30 µL of 1.25% AlCl_3_ were added. After 5 min of incubation, 30 µL of 0.125 M C_2_H_3_NaO_2_ was added. The resultant yellow hue was detected at 420 nm after the incubation period. Data are displayed with standard deviations.

#### 2.5.2. Determination of Antioxidant Activity

With a few minor carried out in microplates, the ABTS assay was based on [28]. A 50 mL volumetric flask had 192 mg of ABTS that had been dissolved in distilled water before the flask was filled with distilled water. After combining 1 mL of the previous solution with 17 µL of 140 mM potassium persulphate, the mixture was left in the dark for 24 h. A total of 1 mL of the reaction mixture was diluted to 50 mL with methanol to provide the final concentration of ABTS for the test. A total of 190 µL of freshly produced ABTS reagent and 10 µL of the sample were mixed in a 96-well plate (*n* = 6). The reaction was then allowed to rest at room temperature in the dark for 120 min. After the incubation, the ABTS color’s intensity at 734 nm decreased. In accordance with Equation (1), data are displayed with standard deviation:(1)$$Percentage\;inhibition\;=\left(\frac{Average\;absorbance\;of\;blank-Average\;absorbance\;of\;the\;test}{Average\;absorbance\;of\;blank}\right)*100$$

For the ferric ion-reducing antioxidant power (FRAP) assay, a newly made tri pyridyltriazine (TPTZ) reagent (300 mM of acetate buffer (pH = 3.6), 10 mM of TPTZ in 40 mM of HCl, and 20 mM of FeCl3, in a ratio of 10:1:1 *v*/*v*/*v*, respectively) was used in this experiment, which was carried out in accordance with the technique in [29], with a few minor changes. A total of 190 µL of freshly prepared TPTZ reagent and 10 µL of the sample were mixed in a 96-well plate (*n* = 3). The reaction was incubated for 30 min at room temperature while kept in the dark. The last measurement of the blue hue after incubation was made at 593 nm. 

The iron metal chelation assay was performed in microplates using the Santos et al. method with a few little modifications [30]. A total of 50 µL of the sample was mixed with 20 µL of freshly made ferrous sulfate (0.3 mM) on a 96-well plate (*n* = 6). Then, 30 L of ferrozine (0.8 mM) was placed in each well. The mixture for the reaction was incubated at room temperature for 10 min. At 562 nm, the resultant color’s intensity was found to have decreased at the conclusion of the incubation period. According to Equation (1), data are shown with standard deviation.

### 2.6. Statistical Analysis 

The experiments were conducted in a completely randomized design with five replicates of both treated and control cultures. The data are shown as means ± standard deviations. GraphPad Prism version 5.01 was used to conduct a t-test for calculation of *p*-value and significance.

## 3. Results and Discussion

### 3.1. Material Properties and Characterization

Figure 1a depicts the morphology of the resulting nanotube structure after the breakdown anodization of the Ti-Mo-Ni alloy sheet. Figure 1b demonstrates the existence of Ti, Mo, Ni, and O elements using EDX, which confirms the retainment of all of the alloy components after breakdown anodization. The XRD spectra in Figure 1c display well-defined diffraction peaks at 2Ѳ = 25.4°, 38.2°, and 53.6° that can be assigned to (101), (004), and (211) TiO_2_ anatase phase planes, respectively. Two diffraction peaks centered at 2Ѳ ≈ 40.3° and 2Ѳ ≈ 62.9° are attributed to the presence of insoluble impurities from Ti metal, which occurred during the breakdown anodization process. Moreover, a distinct diffraction peak is noticed at 35.2° and corresponds to the NiO (111) plane. Another peak was obtained at 48.2°, which can be related to the Ti_2_Ni (442) plane. The tiny diffraction peak positioned close to 42° can be ascribed to the Ni-Mo phase. Importantly, no separate peaks are reported for Mo oxides, suggesting that their ions are incorporated into the TiO_2_ anatase structure.

### 3.2. Ti-Mo-Ni-O NTs Experiment on Philodendron erubescens 

Figure 2 detects the effect of 5.0 mg/L of Ti-Mo-Ni-O NTs powder used with cultured in vitro shootlets of Philodendron that include three treatments (a, b, and c, respectively). (a) This image suggests that the original culture of Philodendron shootlets cultivated on MS solid medium included 0.5 mg/L of BA and Kinetin, 3% sucrose, and 8.0 g/L of agar with a pH of 5.7. (b) This image proposes Philodendron shootlets (as a control) cultivated in liquid MS medium (3% sucrose, 0.5 mg/L of BA, and Kinetin with a pH of 5.2) at 90 rpm for three days without powder of NTs. (c) This image illustrates Philodendron shootlets (as a sample) cultivated in liquid MS medium with 5.0 mg/L of Ti-Mo-Ni-O NTs powder, 3% sucrose, 0.5 mg/L of BA, and Kinetin with a pH of 5.2 at 90 rpm for three days. 

In this study, the NTs used were based on different metals and metal oxides, such as Ti-Mo-Ni-O NTs that have been applied in the cultivation of Philodendron shootlets in vitro, along with an increase in vegetative growth that was observed in treatment (c) after three days compared to treatment (b) as a control. It is believed that nanoparticles (Ti-Mo-Ni-O NTs) led to a decrease in nutrient losses due to leaching and prevented chemical modifications, improving nutrient usage efficiency and environmental quality. Therefore, the nanoparticles caused an increase in the absorption of water, resulting in a quicker rate of seedling growth. In this regard, ref. [5] proved that *Thymus daenensis* seed germination in vitro, seedling development metrics, and secondary metabolite synthesis are all affected by various concentrations of multi-walled carbon nanotubes (MWCNTs), reaching 0, 125, 250, 500, 1000, and 2000 µg mL^−1^, added to MS media. Leopold et al. [31] applied TiO_2_ and ZnO nanoparticles in large quantities to in vitro-grown soybean plants. In most cases, the production of different plant metabolites produced overall patterns for ZnO and TiO_2_ nanoparticles that were similar. Total protein levels were increased by both NP treatments; nonetheless, ZnO NPs have a greater impact than TiO_2_. Similar results were seen for the effects of ZnO NPs and TiO_2_ on the synthesis of chlorophyll, as well as the concentration of carotenoids and chlorophylls. The effect was shown to be dosage-dependent, and both forms of NPs exhibited similar behavior. 

In a recent study [32], the effects of seed priming with TiO_2_ nano-particles on the germination and physiological development of maize exposed to salinity stress were studied [33]. The germination and development of maize seedlings under salinity stress were found to be positively affected by a 60 ppm TiO_2_ nano-priming treatment. Additionally [33,34,35], proper compositions and fabrication methods for nano-fertilizers are crucial in modern agriculture for maximizing the fertilizer’s effectiveness in plants. Nano-fertilizers are able to promote nutrient uptake by roots due to their reduced size (obtained through physical/chemical techniques), which increases their surface mass proportion. In addition, nano-fertilizers are able to promote nutrient uptake by roots because their reduced size (obtained through physical/chemical techniques) increases their surface mass proportion [36]. 

### 3.3. High-Performance Liquid Chromatography with a Diode-Array Detector (HPLC–DAD) Analysis 

Figure 3 and Figure 4 illustrate the HPLC analysis of the Philodendron shootlets extract (without and with) Ti-Mo-Ni-ONTs powder. Moreover, the results demonstrate that rutin as a phytochemical compound from extracts (control and sample) displayed separate peaks at the retention time (minutes), 49.178, and 49.140 min, respectively. Ti-Mo-Ni-O NTs powder extremely increased the amounts of rutin compound in the Philodendron sample to reach 2.96 (mg/g) compared to the control, reaching 0.02 (mg/g), as shown in Table 1.

Secondary metabolites (such as rutin or other phenolic compound) are produced by plants in response to biotic and abiotic elicitors [37]. Nano-elicitation has developed as a new and effective approach for boosting the generation of secondary metabolites. Research on nano-elicitors utilizing various types of nanoparticles, particularly metal oxide nanoparticles, has significantly increased in recent years, driven by the substantial rise in medically significant secondary metabolites. Metal–organic nanoparticles have attracted significant interest because of their distinct physiochemical properties [38]. Thus, Ti-Mo-Ni-O NTs powder significantly increased the rutin concentration in Philodendron shootlet extract by about 148 times compared to the control. The administration of multi-walled carbon nanotubes (MWCNTs) to *Thymus daenensis* seedlings at modest concentrations (250 g mL^−1^) stimulated the development of phytochemical improvements, such as total phenolic compounds and free radical scavenging activities in vitro [6]. 

### 3.4. Biochemical Analysis of Total Phenolic and Total Flavonoid Compounds 

Table 2 shows that Ti-Mo-Ni-O NTs powder significantly increased (*p* ≤ 0.05) the values of the total phenolic content and total flavonoid content of the Philodendron sample, reaching 61.30 ± 2.67 (mg/g) and 29.31 ± 2.62 (mg/g), respectively, compared with the control, reaching 18.56 ± 1.37 (mg/g) and zero (mg/g), respectively. The results display that Ti-Mo-Ni-O NTs powder led to a doubling of the total phenolic content and total flavonoid content, about 2–3 times more than the control. 

The total phenolic content of *Thymus daenensis celak* seedlings’ aerial portions increased sharply after being exposed to multi-walled carbon nanotubes (MWCNTs) [6]. Overall, the findings demonstrate that 250 g mL^−1^ of MWCNT treatment resulted in the highest total phenolic content (6.700 mg/g) and total flavonoid content (8.189 mg/g). The total phenolic content and total flavonoid content increased by 2 and 1.09 times, respectively, compared to the control.

In plants, the production of secondary metabolites (such as total phenolic content and total flavonoid content) can be prompted by exposure to biotic and abiotic elicitors [38]. Carbon nanotubes’ adhesion forces are proportional to the cell wall’s nitrogen content, causing them to interact with the plant’s cellular membrane [39]. Afterward, their filamentous structures penetrate the plant cell wall once they have encased and attached to the surface of the cell [40]. Obviously, carbon nanotubes are elicitors that can affect pathways of signal transduction, expression levels, and the generating enzymes and proteins, all of which impact the generation of secondary metabolites in plants [40]. 

### 3.5. Total Antioxidant Activity Assay 

Table 3 shows that the antioxidant activity of an extract of Philodendron shootlets was evaluated using the ABTS assay, which is a radical cation [2,29-azinobis-(3-ethylbenzothiazoline-6-sulfonic acid)], an electron transfer-based assay. The results display that the extract of Philodendron shootlets (Sample) provided an extremely significant (*p* ≤ 0.05) increase in inhibiting the free radical ABTS^+^ and converting it to the more stable ABTS state through the inhibitory activity of the radical of ABTS^+^ and was estimated to be equivalent to Trolox. The values obtained were 274.55 ± 13.56 (µM TE/mg of extract) compared to the control, reaching 70.23 ± 4.30 (µM TE/mg of extract).

This explains the creation of the blue/green ABTS^+^ chromophore via the reaction of ABTS with potassium persulfate, with maximum absorbance at 734 nm. Therefore, when antioxidants are added to a radical cation after it has already been generated, the ABTS of the cation is reduced; however, the exact amount and rate of reduction depends on the antioxidant’s activity, concentration, and the timescale over which the reaction proceeds. The percentage inhibition of the ABTS^+^ radical cation is used to quantify the degree of decolorization as a function of concentration and duration, with results normalized to the reactivity of Trolox under identical experimental conditions [28,41].

On the other hand, Table 4 shows that the antioxidant activity of an extract of Philodendron shootlets was evaluated using a ferric-reducing antioxidant power (FRAP) assay. The results illustrate that Philodendron shootlet extract (Sample) provides a significantly (*p* ≤ 0.05) increased reduction of Fe(TPTZ)2(III), converting it to Fe (TPTZ)2(II). The reduction potential of Fe(TPTZ)2(III) was expressed to be equivalent to Trolox, reaching 106.62 ± 7.13 (µM TE/mg of extract) compared to the control, reaching 8.70 ± 0.88 (µM TE/mg of extract).

The FRAP test requires the presence of a species of a single electron donor or antioxidants that have a reduction potential of less than 0.7 V and a pH of 3.6 to change the pale yellow reagent of Fe-tri-pyridyltriazine Fe(TPTZ)2(III) to the blue color of Fe (TPTZ)2(II). To measure this reduction, the variation in absorption at 593 nm was investigated [41,42].

Table 5 shows that the antioxidant activity of an extract of Philodendron shootlets was evaluated using the iron metal chelation assay. The results proved that the extract of Philodendron shootlets (Sample) yields an extremely significant (*p* ≤ 0.05) increase in the inhibition of the complex formation of ferrozine–Fe^++^, which was estimated to be equivalent to EDTA, and the value obtained was 11.71 ± 0.89 (µM EDTA eq/mg of extract) compared to the control, reaching 9.41 ± 0.57 (µM EDTA eq/mg of extract). The presence of transition metals, notably Iron(II), in living systems (plants and herbs) raises concerns that they may function as pro-oxidants. A pro-oxidant is not directly harmful to biomolecules, but it does help generate potentially damaging species that can.

In this regard, the ferrozine assay was used to calculate the amount of chelated ferrous ion (Fe^++^) by plant/herb extracts. The formation of a compound between ferrozine and Iron(II) is detectable by spectrophotometry at 562 nm. Reportedly, certain polyphenolic chemicals prevent this complex from forming, leading to lower absorbance [41,43]. 

The outcomes achieved by using HPLC analysis, the total phenolic content and total flavonoid content, ABTS radical cation, the FRAP, and the iron metal chelating activity assays are indicative that Philodendron shootlet extract treated with Ti-Mo-Ni-O NTs powder led to an increase in the sources of a natural antioxidant, such as rutin, with good radical-scavenging activities. Keeping in view the total antioxidant and radical-scavenging activity of the Philodendron shootlet extract treated with Ti-Mo-Ni-O NTs powder comparable to Philodendron shootlet extract without treated Ti-Mo-Ni-O NTs powder as a control may be used against chronic diseases and in other food industries in future.

Nanomaterials have the ability to protect plants from the detrimental impacts of abiotic stressors by activating their antioxidative defense system. They have more effective adsorption and targeted delivery due to their ability to penetrate plants and large surface areas and can be responsible for regulating photosynthetic efficiency and water uptake, as well as detoxifying reactive oxygen species, thereby enhancing the seed germination, growth, and yield of crops. They can be used responsibly in agriculture for greater production by carefully analyzing the dose to be used for different nanomaterials. Nano-fertilizers (NFs) considerably increase plant growth efficiency, soil quality, and crop yield of high-quality fruits and cereals. Globally, managing macro–micronutrients is a challenging issue since it largely relies on synthetic chemical fertilizers, which may not be healthy for human beings or the environment and may be costly for farmers. By controlling the availability of fertilizers in the rhizosphere, NFs may strengthen plant defense systems, increase nutrient absorption, and prolong the capability of plants to withstand stress [37]. Plants produce phytochemicals as secondary metabolites to protect themselves from environmental threats, such as microbial diseases, pollution, temperature swings, and drought. Rutin is a flavonoid phytochemical that possesses potent antioxidant properties. It has significant positive benefits on human health, including antioxidant, capillary fragility therapy, anti-Alzheimer, and anti-inflammatory properties [44]. In our study, we show an extremely significant increase in rutin content compared to the control.

## 4. Conclusions

The use of mixed metal oxide nanotubes for the first time in plant nano-biotechnology showed tremendous performance as a novel elicitor. Ti-Mo-Ni-O NTs extremely increased the rutin active compound in Philodendron “Imperial Red” (*Philodendron erubescens*) in vitro plantlets. The rutin content increased in the sample as compared to the control from 0.02 mg/g to 2.96 mg/g for dry-weight shootlet extracts. This superior increase, 148 times larger after the addition of 5 mg of Ti-Mo-Ni-O NTs, paves the way for the use of metal oxide nanotubes with other plant species. 

## Data Availability

The corresponding authors will provide the datasets used and/or analyzed for the current work upon reasonable request.

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
