# Peer review of "In Vitro: The Extraordinary Enhancement in Rutin Accumulation and Antioxidant Activity in Philodendron “Imperial Red” Plantlets Using Ti-Mo-Ni-O Nanotubes as a Novel Elicitor"

_biotech, 2024, doi:10.3390/biotech13030024_

Round 1
Reviewer 1 Report
Comments and Suggestions for Authors
The manuscript needs to be improved in many aspects. Some of them are highlighted here.
#1. Line 58- Authors mentioned “Titanium (Ti) is a beneficial element for plants”, is it so? but next statement mentioned when it is in low concentration. Its better to provide a specific statement instead of general statement.
#2. Similarly line 70, “Mo is necessary for the growth of most plants”- is it necessary?
#3. Line 88-“synthesis of irritating waste products-“- irritating?
#4. Line 89- “Therefore, stresses the biogenic”- No meaning
#5. Mention the organism’s name in italics
#6. What is the yield of Ti-Mo-Ni Oxide NTs?
#7. Why specific About 5.0 mg/L of Ti-Mo-Ni Oxide NTs are used for experiement? Why not different concentrations are tested?
#8. “Cultured shoots were shaking at 90 rpm for three days”- How come with in 3 days the rutin content will be significantly increased?
#9. In that case what is the changes of other growth parameters of plant? Why those results are not recorded and reported?
#10. What is the statistical inference? What is the technical or biological replicates?
#11. Why only 3 days incubation is employed in this study? Why no variables tested?
#12. Line 153, “one milligram of”- in other places it mentioned as unit like mg or ug. Then why its mentioned here in words. Maintain same format.
#13. Figure 1 a contains a inset image , which is not clear
#14. Line 227, “while (c) suggesting”- can be modified with proper sentence
#15. Line 245, “Leopold et al. 36 Demonstrated” check it
#16. Line 269, “Figures 3 and 4 illustrate the HPLC analysis of the Philodendron shootlets extract that was (without and with) Ti-Mo-Ni-ONTs powder”- No proper meaning
#17. How the specific peak in HPLC spectrum is identified as rutin ?
#18. Provide the reference for the statement provided in the line 314-316
#19. Line 320, “The results display the extract of Philodendron shootlets (Sample) has an extremely significant (p ≤ 0.05) increase? Did the extract has an increase or the shootlets has an increase. Provide a proper meaningful sentence
#20. Results are compared in the basis of principles of the experiments carried out. However, results are not compared with the results of previously published related papers
#21. There are so many scientific gaps in the result interpretation, which need to be changed.
#22. Many sentences are not giving proper meaning. So its better to read the manuscript again and revise the junk sentences
#23. Language proof reading is required
#24. Reference 2, 22,23,24,25, 33,34,35 self-citations are not too much relevant to be cited here in this work
Comments on the Quality of English Language
The English language proof reading might be required.
Author Response
Thank you very much your comments

Reviewer 2 Report
Comments and Suggestions for Authors
The presented article, " In Vitro: The Extraordinary Enhancement in Rutin Accumulation, Antioxidant Activity in Philodendron “Imperial Red” Plantlets using Ti-Mo-Ni-O Nanotubes as a Novel Elicitor" is written at a commendable scientific level and proves to be quite engaging for readers.
In their manuscript, the authors studied Philodendron erubescens shoots grown in vitro for the accumulation of Rutin compounds using Ti-Mo-Ni-O NTs powder, then determined the quantitative and qualitative accumulation of Rutin by (HPLC–DAD) analysis and estimated the antioxidant activity for their Philodendron erubescens shoot extract. This study presents a novel methodology for utilizing nanobiotechnology in the synthesis of plant secondary metabolites.
The authors used appropriate methodology and experimental design to test the study hypothesis, also.
I think that the manuscript’s results could be reproduced based on the details given in the methods section. It is an important work, which could be helpful to researchers.
The paper is well organized, easy readable and presented in a well-structured manner.
Therefore, I recommend that the authors address the following aspects to enhance the quality of their study:
1. Abstract - P. erubescens – this abbreviation must be specified from the beginning.
2. Lines 40-42: Please use another reference and remove reference 2 (auto citation); it is not relevant for this paragraph.
3. Lines 45-50: Please add the references.
4. Lines 58-60: Please add the references.
5. Lines 101-103 – Please remove reference 23. The references 22 and 24 can remain and in addition could be added other references, studies by other researchers, can be used.
6. Line 138 – Please remove the reference 27 (auto citation) – it is not relevant.
7. Revise lines: 161, 245, 272 (spaces should be added between some words or the capital letter should be corrected with the small letter at the beginning of the word).
8. Lines 221-229 - too long paragraph – could be rewritten.
9. Lines 255-257 - the paragraph should be rewritten.
10. Line 277-279 – please revise! The increase from 0.02 to 2.96 (mg/g) represents a 104 times increase?
11. Line 279: The abbreviation "MWCNT" must be specified.
12. Figure 1 has too low dimensions. It is too difficult to see details. It could be split into 2 or even 3 independent figures, so that the images could be more visible.
13. Could you please provide information on the sizes of nanotubes used?
14. What is the reproducibility of nanotubes synthesis in terms of size and properties?
15. Could the authors suggest a possible mechanism by which tested nanotubes manage to activate the Rutin level in this way?
16. Discussions paragraphs must be improved and used much more references.
17. Consider expanding the discussion on the potential applications of these nanotubes in future.
Author Response
Thank you very much for your comment

Round 2
Reviewer 1 Report
Comments and Suggestions for Authors
Line 88-“synthesis of irritating waste products-“-The word irritating means? clarification provided by the authors is not justified
The mass of the alloy sheet with the dimensions of 1 mm × 1 cm × 1 cm is ≈2 grams.- How it is measured and the analytical results are not provided
“Cultured shoots were shaking at 90 rpm for three days”- How come within 3 days the rutin content will be significantly increased?- No proper justification is provided
What is the statistical inference? What is the technical or biological replicates? Authors mentioned this is Not unerstood- It is been asked how many times the experiment performed at same time and different time interval.
The manuscript still need a improvement in the writing part and providing supporting results
Comments on the Quality of English LanguageLanguage revision is required, most sentences are not providing the straight meaning.
Author Response
Thank you very much for your comments, we have replied all comments (a point-by-point) in the attached file and in the manuscript as track change
Thanks in advance

Round 3
Reviewer 1 Report
Comments and Suggestions for Authors
Nil